# Effect of Post-Processing Heat Treatments on Short-Term Creep Response at 650 °C for a Ti-6Al-4V Alloy Produced by Additive Manufacturing

Chiara Paoletti [1,*], Marcello Cabibbo [2], Eleonora Santecchia [2], Emanuela Cerri [3] and Stefano Spigarelli [2]

1   Faculty of Engineering, Università degli Studi eCampus, Via Isimbardi 10, 22060 Novedrate, Italy
2   DIISM, Università Politecnica delle Marche, Via Brecce Bianche, 60131 Ancona, Italy;
    m.cabibbo@univpm.it (M.C.); e.santecchia@univpm.it (E.S.); s.spigarelli@univpm.it (S.S.)
3   DIA, Università di Parma, V. le G. Usberti 181/A, 43124 Parma, Italy; emanuela.cerri@unipr.it
*   Correspondence: chiara.paoletti@uniecampus.it

**Abstract:** Post-processing heat treatments of Ti-6Al-4V parts produced by additive manufacturing are essential for restoring the peculiar martensitic structure that originates from the extremely high cooling rates typical of this technology. In this study, the influence of a 1050 °C annealing on a Ti-6Al-4V alloy, produced by additive manufacturing, on the minimum creep rate dependence on applied stress and temperature, was investigated at 650 °C. Experimental data obtained after two different subcritical annealings were also considered for comparison purposes. The analysis of the experimental creep data demonstrated that the alloy annealed at the highest temperature exhibited lower creep rates. The improved creep response was attributed to the combined effect of the presence of extended $\alpha$-$\beta$ interfaces and of a small volume fraction of $Ti_3Al$ particles.

**Keywords:** creep; titanium alloys; additive manufacturing





## 1. Introduction

The creep response of Ti-6Al-4V alloy produced by additive manufacturing is a subject that deserves new investigations, due to the peculiarities of the process. The extremely high cooling rates (up to $10^6$ °C/s) produce a unique microstructure, since the high-temperature bcc $\beta$-phase massively decomposes by a martensitic (diffusionless) transformation. As a result, the microstructure of the as-built part is fully composed by $\alpha'$-martensite, and for this reason invariably necessitates a subsequent post-processing annealing treatment. A very recent study [1] indeed analyzed the creep response of the annealed Ti-6Al-4V produced by selective laser melting (SLM), a technology also known as laser beam powder bed fusion (LB-PBF), placing it in a wider context, that of the effect on creep of the initial microstructures of this alloy. The $\alpha$-$\beta$ phase transformation in Ti-6Al-4V, and the resulting volume fractions, distributions and morphologies of the constituent phases, strongly depend on the temperature history and cooling rates. The same alloy can thus exhibit equiaxed, Widmanstätten or even duplex microstructures because of different combinations of thermomechanical processing, heat treatment temperatures and cooling rates. A large set of experimental results obtained by testing equiaxed [2–7], duplex [8,9] and Widmanstätten [2,10–13] microstructures was thus reviewed in [1]. The analysis led to a constitutive model that was perfectly adequate in describing the creep data obtained by testing the alloy produced by SLM in the as-deposited state [14] and after annealing at 740 °C (the original data presented in [1]). The minimum creep rate ($\dot{\varepsilon}_m$) dependence on applied stress ($\sigma$) and temperature ($T$) was well described by a unique constitutive equation, in the form

$$\dot{\varepsilon}_m = A \frac{D_0 G b}{kT} \left( \frac{\sigma_\rho}{G} \right)^3 exp \left( \frac{\sigma_\rho b^3}{kT} \right) exp \left\{ -\frac{Q_L}{RT} \left[ 1 - \left( \frac{\sigma_\rho}{R_{max}} \right)^2 \right] \right\} \qquad (1)$$

where $G$ was the shear modulus, $k$ the Boltzmann constant, $b$ the Burgers vector, $R$ the gas constant, $A = 40$, and $Q_L = 303$ kJ mol$^{-1}$ and $D_0 = 1.0 \times 10^{-3}$ m$^2$s$^{-1}$ were the activation energy for self-diffusion in Ti and the relevant material constant in the Arrhenius form equation for mass diffusion [15]. Lastly,

$$\sigma_\rho = (1 - \delta)\sigma \tag{2}$$

with $\delta = 0.4$. The $\delta\sigma$ term was intended to represent the strengthening effect of solute atoms due to viscous glide of dislocations. The $R_{max}$ term was calculated as

$$R_{max} = 1.5 UTS \frac{G_T}{G_{RT}} \tag{3}$$

where $UTS$ is the ultimate tensile strength at room temperature ($RT$), with $G_T$ and $G_{RT}$ being the shear modulus values at the investigated $T$ and at $RT$.

The most intriguing finding in [1] was that, in a wide range of temperatures and applied stresses, Equations (1)–(3) were perfectly capable of describing all the reviewed sets of experimental data, irrespective of the initial microstructure and production technology, with the partial exception of the alloys with Widmanstätten microstructure. The creep results obtained by testing the alloy with this microstructure, usually produced by an annealing above $\beta$-transus, in fact, seemed to be affected by a remarkable experimental scatter, since the minimum creep rates at a given temperature formed a sort of cloud, spanning over orders of magnitude, rather than collapsing on a single curve. In this context, the possible role of an annealing above the $\beta$-transus, when applied to a material produced by additive manufacturing, becomes an intriguing problem. The aim of the present study was thus to elucidate the effect of a high-temperature annealing on the minimum creep rate dependence on applied stress and temperature, on the same SLM-produced alloy tested in [1]. For comparison purposes, also experimental data obtained after two different subcritical annealings were considered. The experimental temperature (650 °C) was purposely selected to minimize the risk of precipitation of an additional phase, Ti$_3$Al ($\alpha_2$), which is commonly thought to form below 600 °C [16].

## 2. Materials and Methods

Ti-6Al-4V powders used in this study had the following chemical composition reported in Table 1 (grade 23 ELI). The commercial gas-atomized powder particles had sizes ranging between 15 and 45 μm. A selective laser melting SLM280 machine (SLM Solutions AG, Lübeck, Germany), equipped with a 400 W IPG fiber laser, produced flat dog-bone creep samples with the shape illustrated in Figure 1.

**Table 1.** Chemical composition of the commercial powders (wt.%).

| Al | V | Fe | O | N | C | H | Ti |
|---|---|---|---|---|---|---|---|
| 5.5–6.5 | 3.5–4.5 | 0.25 | 0.1 | <0.05 | <0.08 | <0.011 | bal. |

The powder bed was pre-heated at 80 °C and purged with argon to reduce oxygen concentration to 0.05%. Building direction was parallel to the gauge length of the specimen. The layer thickness was 60 μm, the scanning speed 1250 mm/s, the hatch spacing 120 μm, and laser power 340 W. Both sample production and post-processing heat treatments were performed by BEAMIT (https://www.beam-it.eu/ (accessed on 22 June 2022), Fornovo di Taro 43045—Italy).

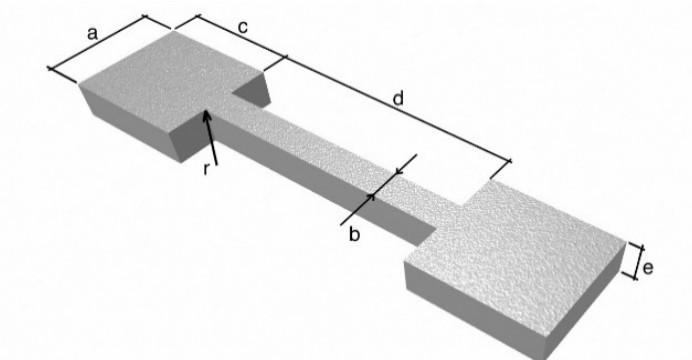

**Figure 1.** Geometry of the creep samples (a = 10 mm, b = 3 mm, c = 15 mm, b = e = 3 mm, and r = 0.5 mm).

Three different post-processing treatments were used to modify the microstructure of the alloy (Figure 2). The first treatment aimed at obtaining a better ductility without significant loss in strength. This treatment consisted of annealing for 130 min at 740 °C, followed by furnace cooling in Ar down to 520 °C (average cooling rate $C_r$ = 0.04 °C/s), followed by a faster cooling ($C_r$ = 0.3 °C/s) to R.T., again in Ar (condition "T1", already investigated in [1]). The second post-processing treatment consisted of annealing at 1050 °C, i.e., well above $\beta$-transus, for 1 h, followed by cooling in Ar (condition "T2", $C_r$ = 0.4 °C/s). Similar treatments are known to cause full recrystallization; in addition, the relatively slow cooling should lead to the formation of a Widmanstätten structure with reduced internal stresses [17]. The last treatment, the "reference" one, consisted of a 1 h annealing at 704 °C, followed by cooling in argon ($C_r$ = 0.35 °C/s), as prescribed by AMS2801 standards (condition "R"). Post-processing heat treatments were carried out in an NADCAP accredited vacuum furnace (TAV, Caravaggio, Italy), which underwent periodic pyrometry testing in line with the AMS 2750 standard.

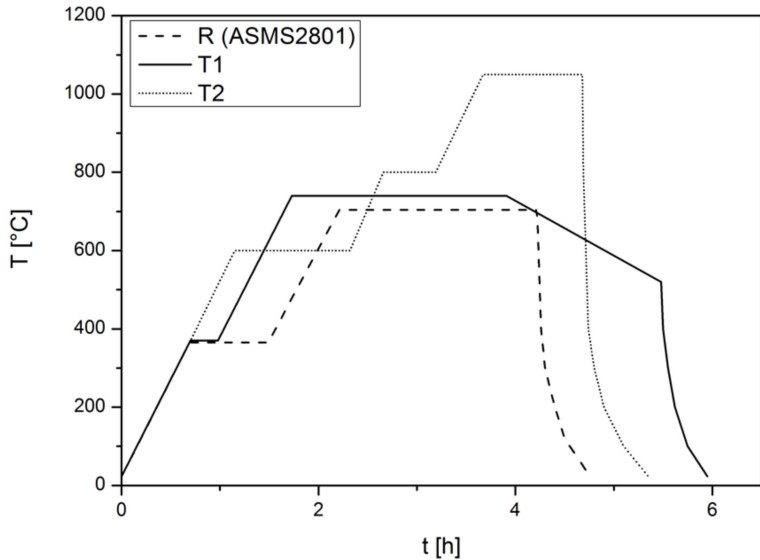

**Figure 2.** Temperature vs. time plots for the investigated post-processing T1 (740 °C), T2 (1050 °C) and R (704 °C) heat treatments.

Constant load creep experiments were carried out at 650 °C in air on samples whose surfaces were in the as-deposited finishing. Initial nominal stresses for samples in T2 state were 60, 100, 120, 150, 180, 270, 330 and 385 MPa. Additional tests were carried out under 100, 275 and 385 MPa on samples in R state. To maintain a homogeneous heating profile in the furnace, the test temperature was measured using four thermocouples. Elongation was continuously measured using a linear variable displacement transducer (LVDT). Loading

occurred after a short soaking at the testing temperature for all tests except one, which was carried out on a sample in T2 state. This single sample experienced a prolonged (145 h) permanence at 650 °C before loading. The experiment was aimed at assessing the possible effects of aging on creep response.

Rockwell hardness (HRC) tests were carried out on the T2 crept-sample heads. Additionally, these tests were aimed at assessing the age-hardening response of the alloy, and for this reason, the hardness was measured in these (almost) unstressed portions of the samples—the hardness of the gauge lengths being significantly influenced by dislocation substructures introduced by loading.

Tensile tests at room temperature were carried out on samples of the geometry presented in Figure 1, with the surface in the as-deposited state to evaluate the UTS. The testing strain rate was $3 \times 10^{-2} \, \text{s}^{-1}$. The UTS values were $1006 \pm 12$ MPa, $1015 \pm 5$ MPa and $922 \pm 2$ MPa for R, T1 and T2 samples, respectively. Initial HRC values were $40.9 \pm 0.3$, $39.8 \pm 0.3$ and $38.1 \pm 0.5$ for R, T1 and T2 samples, respectively.

Samples for light microscopy were mechanically ground and polished with a colloidal suspension and etched by Kroll's reagent (100 mL $H_2O$ + 2 mL HF + 4 mL $HNO_3$). Samples were then characterized by a Leica DMi8 (Leica Microsystems, Wetzlar, Germany) optical microscope equipped with an image analyzer that was preliminarily used to estimate the volume fraction of porosity in the as-deposited samples (1%).

Scanning electron microscopy (SEM) was used for the fractographic analysis of the crept rupture surface. The observations were carried out with a Tescan Vega 3 (Tescan, Brno, Czech Republic) scanning electron microscope using an accelerating voltage of 30 KeV.

Thin foils for transmission electron microscopy (TEM) were obtained from untested and crept sample heads. The samples were prepared by mechanical grinding and polishing down to a thickness of 150 μm. A further thickness reduction down to 60 μm was obtained by twin-jet electro-polishing using a Struers™ (Struers Inc., Westlake, Cleveland, OH, USA) Tenupol-5® device with a solution consisting of 5% perchloric acid, 35% butanol, and 60% methanol at −35 °C and a voltage V = 24 V. The 60 μm thick discs were then dimpled down to a thickness of 20–25 μm and ion-milled to electron transparency using a Gatan© (Gatan Inc., Pleasanton, CA, USA) PIPS working at 8 keV (incident beam angle progressively reducing to 8, 6, and 4°). TEM inspections were carried out by a Philips (Philips It, Milano, Italy) CM-20 microscope working at 200 kV and equipped with a double-tilt LN cooled specimen holder. Selected area diffraction patterns (SAEDP) were recorded by converged beam. Crystallographic structures of $\alpha$, $\alpha'$, $\alpha_2$ (Ti$_3$Al) were inspected by SAEDP acquired as to have $\alpha$-[0001], or $\alpha$-[2$\bar{1}\bar{1}$0] parallel to the beam direction.

### 3. Results

*3.1. Initial Microstructure*

Figure 3 shows the initial microstructure after T1 and T2 heat treatments, as observed by optical microscopy. The analysis of Figure 3a revealed the typical microstructure of an alloy produced by SLM. Optical microscopy cannot properly distinguish between $\alpha$-phase and $\alpha'$-martensite, but the previous study carried out on the samples annealed at 740 °C [1] yielded precious indications on the nature of this microstructure. The martensitic $\alpha'$ microstructure of the additively manufactured Ti-6Al-4V, as widely documented in [1], was only partially decomposed in the $\alpha + \beta$ phases after T1 treatment. The microstructure of the samples in that state thus consisted of 90% of $\alpha$-phase, and of an equal volume fraction of $\alpha'$-martensite and $\beta$-phase. Figure 3b,c show the typical morphology of the alloy annealed above $\beta$-transus, and cooled with a relatively low cooling rate. The Widmanstätten microstructure is well evident, and a thin layer of light $\alpha$-phase decorates prior-$\beta$ grains.

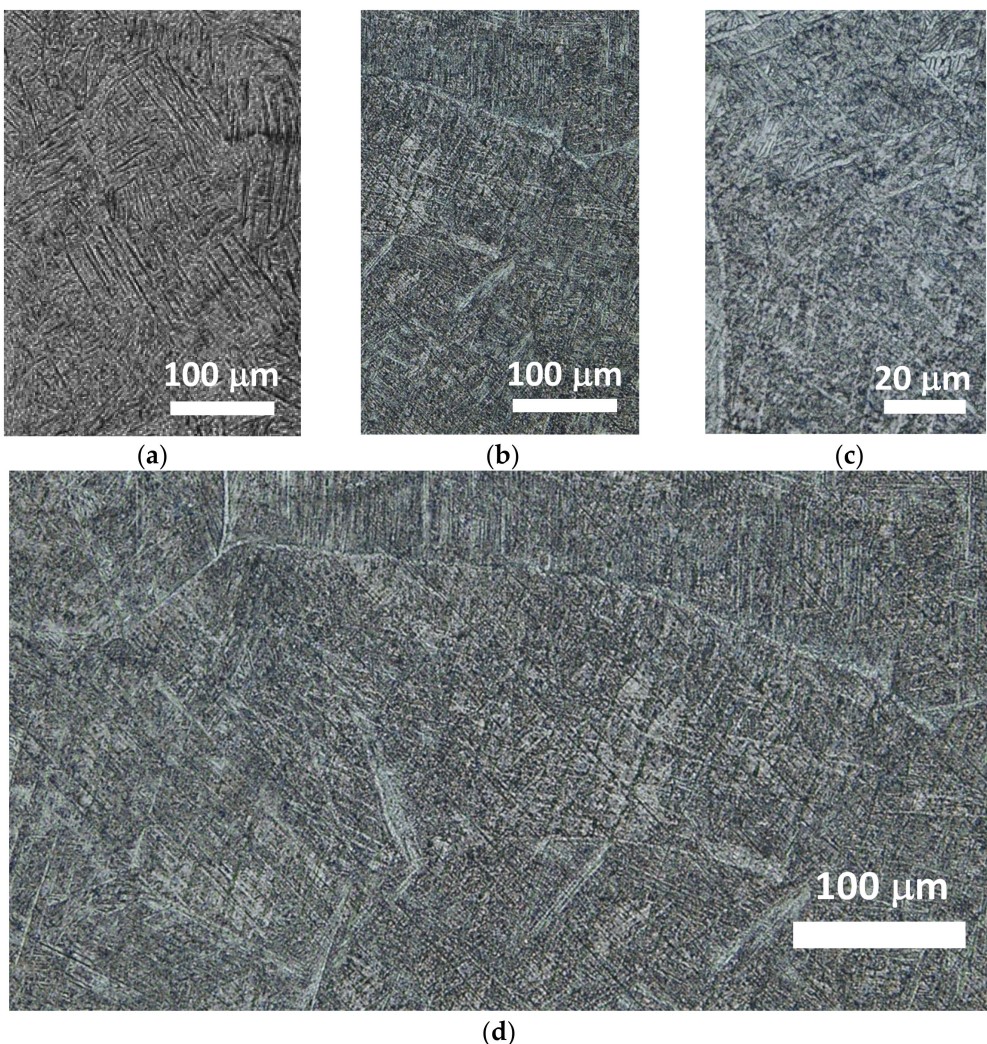

**Figure 3.** Optical micrographs of the heat-treated Ti-6Al-4V produced by SLM: (**a**) T1 condition; (**b**,**c**) T2 condition; (**d**) is an enlargement of (**b**).

TEM representative micrographs in Figure 4 illustrate the microstructure of the sample after T2 treatment. The low-magnification micrograph of Figure 4a shows a bi-lamellar structure consisting of fine and long $\beta$-lamellae interposed between coarser $\alpha$-lamellae. The microstructure of the T2 initial state has thus a Widmanstätten morphology resulting from formation of $\alpha$ phases along prior $\beta$ grain boundaries, with colonies of lath-type $\alpha'$-martensite within $\alpha$ phase. The thicknesses of $\beta$ and $\alpha$ lamellae were in the range 110–140 and 390–460 nm, respectively. Hence, the lateral width of $\beta$-colonies was approximately four times lower than that of the $\alpha$ ones. On the other hand, $\alpha'$ martensite colonies have essentially the same thickness of the $\alpha$ lamellae, and were evenly distributed in between the $\alpha$ phase of the Widmanstätten microstructure. In addition, $\alpha'$ appeared to be significantly more equiaxed than the $\alpha$ phase.

A further important microstructural feature was the presence of tangled dislocations inside $\alpha'$ phase. These were observed to slide preferentially along the $\alpha'$-(0002) crystallographic direction, as reported in Figure 4b. A noticeable fraction of the structure was indeed constituted by $\alpha'$ martensite, as also reported by other authors [18–20].

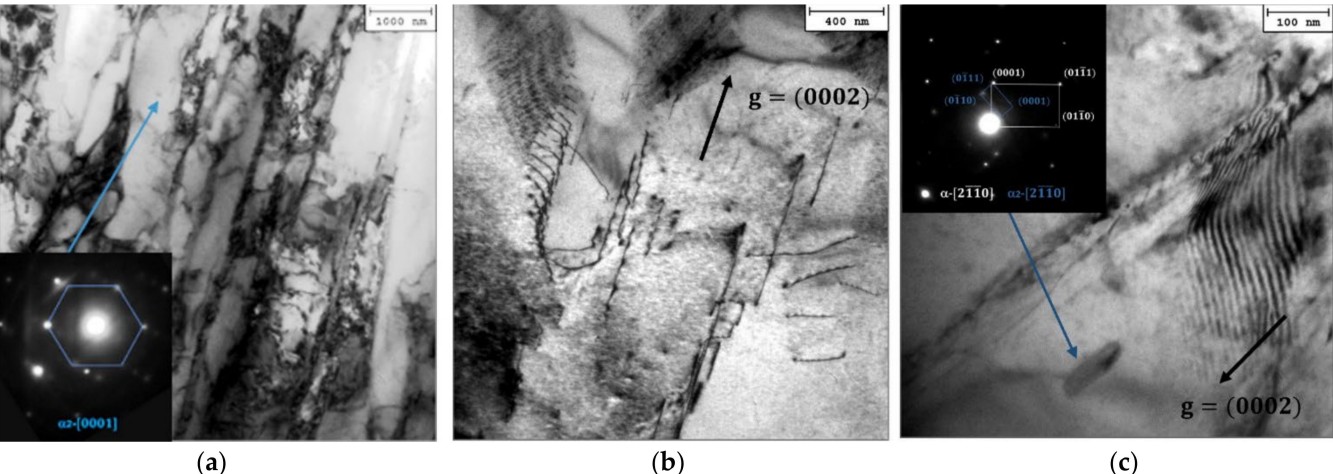

**Figure 4.** Representative TEM micrographs of the initial microstructure. In (**a**) typical lamellar structure of $\alpha$, $\alpha'$, and $\beta$ is reported, with the inset showing SAEDP of $\alpha$ in its [0001]-zone axis; (**b**,**c**) refer to the same $\alpha$-[$2\overline{1}\overline{1}0$]-zone axis. In (**c**), the inset shows indexed SAEDP taken at the $\alpha/\alpha_2$ interface.

Another important microstructural finding is the—somewhat unexpected—presence of $\alpha_2$ Ti$_3$Al phase particles. This phase had an elliptical morphology with typically 90-to-120 nm length and 35–50 nm width. This precipitation of $\alpha_2$-Ti$_3$Al having DO$_{19}$ crystallographic structure, presumably occurred during cooling from annealing temperature, since this phase was not observed after T1 treatment. Ti$_3$Al formed within the hcp-$\alpha$ phase through a short-range rearrangement of the Ti and Al atoms that allowed the crystallographic structure of this phase to remain quite similar to the parent hcp phase. Indeed, only the cell size was changed, to essentially double in $\alpha_2$ relative to that of $\alpha$ phase [21–25]. A representative example for this phase is shown in Figure 4c. Systematic analyses of the crystallographic relationship between the $\alpha_2$ and $\alpha$ phases allowed us to determine that mostly $\alpha_2$ formed with its $\left[2\overline{1}\overline{1}0\right]$ zone axis parallel to the same $\alpha$-$\left[2\overline{1}\overline{1}0\right]$ zone axis, with $\alpha_2$-$(0\overline{1}11)$ direction parallel to $\alpha$-$(0001)$ crystallographic direction.

*3.2. Creep Response*

Figures 5 and 6 show representative strain vs. strain rate curves obtained at 650 °C for the investigated conditions. Figure 5, in particular, clearly shows that the shape of the creep curves was substantially similar for the three different initial states, and consisted of a well-defined primary region, a minimum creep rate range, and a pronounced tertiary stage.

Figure 6, on the other hand, reports the curves obtained under the same stress (100 MPa). The plot clearly shows that the curves obtained after heat-treating the alloy at 704 °C and 740 °C completely overlap. By contrast, the sample heat-treated above the $\beta$-transus exhibited lower creep rates.

This evidence was fully confirmed when plotting the minimum strain rate as a function of initial applied stress (Figure 7, which also shows two curves, whose genesis will be described in a following Section). Again, the data obtained by testing the samples in T1 and R states overlapped on the same curve, while the minimum creep rates for the T2 samples were substantially lower, in particular in the low-stress region. For comparison purposes, Figure 6 also plots the data obtained in [14] by testing another alloy produced by SLM in as-deposited state. In this case, the minimum creep rates were closer (actually a bit higher) than those for the annealed state, which implies that $\alpha'$-martensite does not creep much differently with respect to equilibrium $\alpha$-phase. Last, but not least, Figure 7 shows another interesting result, since, in the case of the T2 state, the minimum creep rate was only marginally affected by a previous long (145 h) permanence at high temperature before loading.

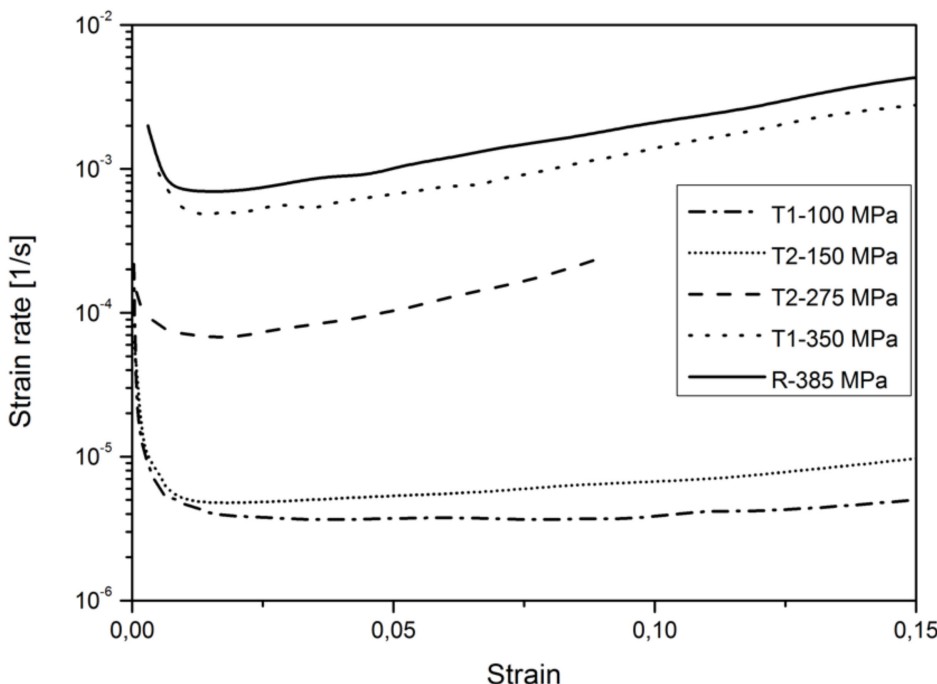

**Figure 5.** Strain rate vs. strain creep curves: representative curves for the three investigated conditions.

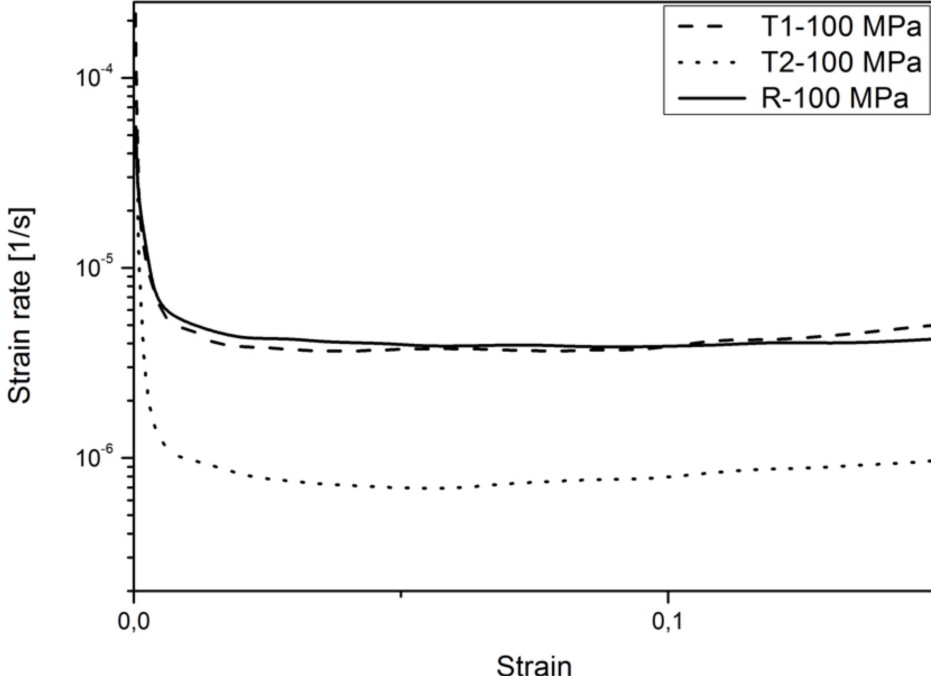

**Figure 6.** Strain rate vs. strain creep curves: comparison between the curves obtained under the same applied stress.

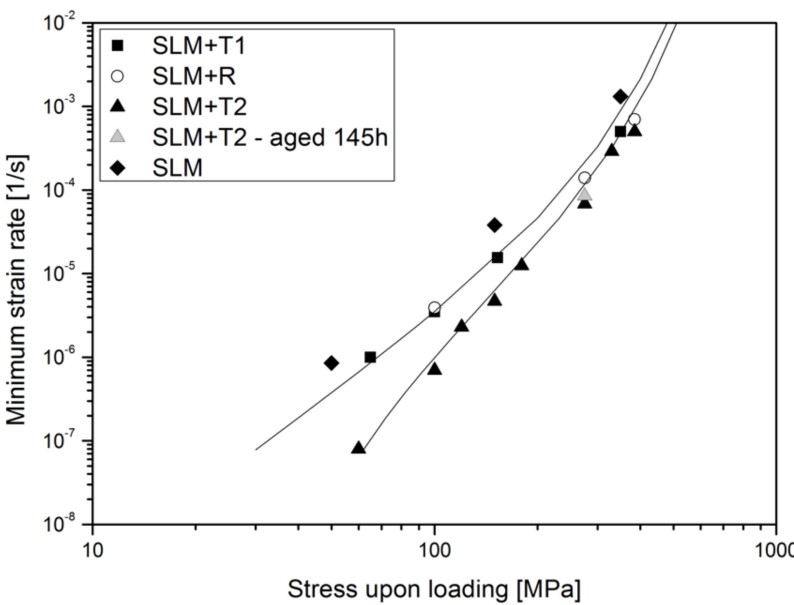

**Figure 7.** Minimum strain rate dependence on applied stress for T1, T2 and R states; for comparison purposes, also data for a similar alloy tested in as-deposited state are reported (data from [14]). The procedure for curve calculation is illustrated in the Section 4.

### 3.3. Hardness Variation in Unstressed Portions of the Samples (Aging Response)

Figure 8 plots the HRC hardness as measured on the sample heads after creep exposure. Hardness slightly increased for very short times at 650 °C, and then slowly decreased with time of exposure. This behavior is representative of a very moderate age-hardening effect, which, as mentioned above, did not have significant implications, at this temperature, on creep response.

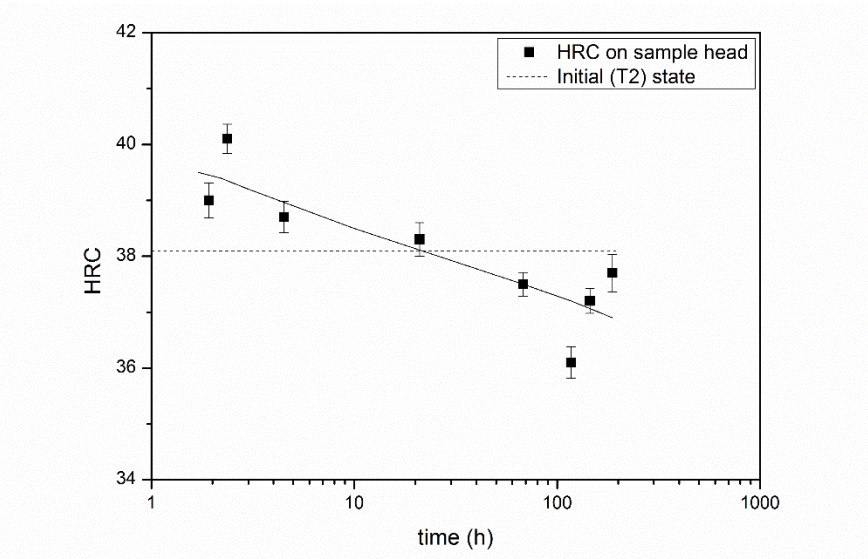

**Figure 8.** HRC variation and standard error with time of exposure. Hardness was measured on the heads of the crept samples to investigate the possible aging/over-aging in almost unstressed portions of the specimen.

### 3.4. Microstructure

The basket-weave morphology microstructure of the alloy exposed at 650 °C, as observed in an almost unstressed portion of the sample (the head), is shown in Figure 9. This structure consisted of coarser lamellar $\alpha$-phase, with some $\alpha'$-martensite colonies,

and needle-like $\beta$-phase. The lateral width of the $\alpha$ phase was larger in this case with respect to the measured width of the initial microstructure (see Figure 4a). The basket-weave structure was characterized by a sequence of $\alpha$ and $\beta$ lamellae meeting at angles in the range of 50–75°. The $\alpha$ and $\alpha'$ phases had parallel geometric orientations and the same grain orientation. At the interfacial region between the $\alpha'$ and $\alpha$ laths, many dislocations were generally formed due to a local strain accommodation followed by both $\beta$-$\alpha$ and $\beta$-$\alpha'$ transformations. Figure 9 also shows some stacking faults (SFs) formed within the $\alpha$ lamellae.

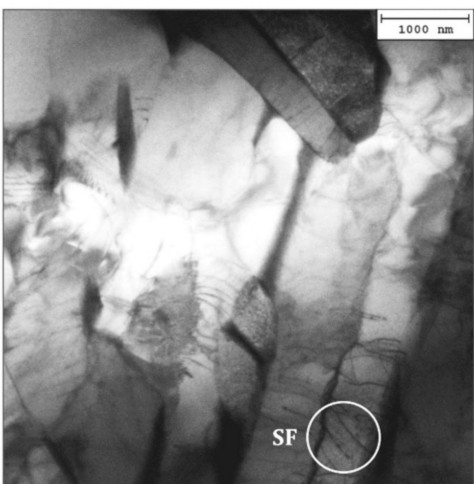

**Figure 9.** TEM micrographs after 187 h at 650 °C showing the basket-weave $\alpha$-$\beta$ lamellar morphology.

The presence of $\alpha_2$-Ti$_3$Al was again reported in the sample exposed at 650 °C (see Figure 10). This phase was found to be significantly coarser with respect to its typical size in the initial sample condition, since Ti$_3$Al particles were typically 120–150 nm wide and 600-to-800 nm long. That is, after high-temperature exposure, the $\alpha_2$ phase coarsened by four-to-five times. The observed significant size increase in the $\alpha_2$ phase particles clearly showed that its formation was driven by a process of nucleation and growth that was induced by short-range atomic diffusion [25].

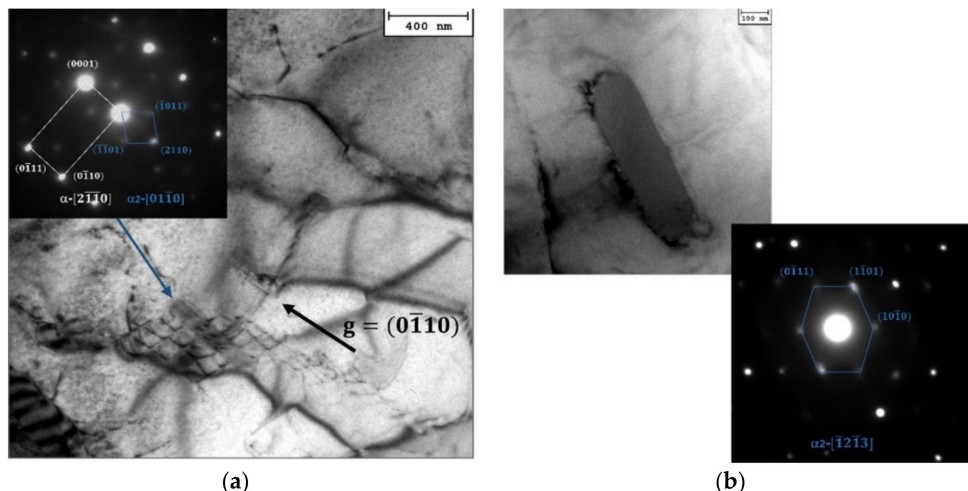

(a)                                        (b)

**Figure 10.** TEM micrographs of the sample head after 187 h at 650 °C showing in (**a**) the presence of $\alpha_2$, whose crystallographic relationship with the parent $\alpha$ phase is reported in the inset SAEDP; (**b**) typical size of the $\alpha_2$ phase particles.

### 3.5. Fractographic Analysis

Figure 11 shows the fracture surface of the crept sample tested under 275 MPa. At low magnifications (Figure 11a), the surface presented a morphology quite typical of the intergranular fracture. However, at higher magnifications (Figure 11b), the grain boundary region presented the dimple-rupture morphology typical of a ductile fracture. The obvious explanation is that a ductile soft phase, located in the grain boundary region, underwent ductile rupture well in advance of the harder grain-interior zone. This picture is fully consistent with the morphology of the microstructure presented in Figure 3d. The very thin layer of softer $\alpha$-phase on the grain boundaries, due to its very low volume fraction, did not significantly influence the strain accumulation and the minimum strain rate, but was critically important in determining the fracture mode of the sample.

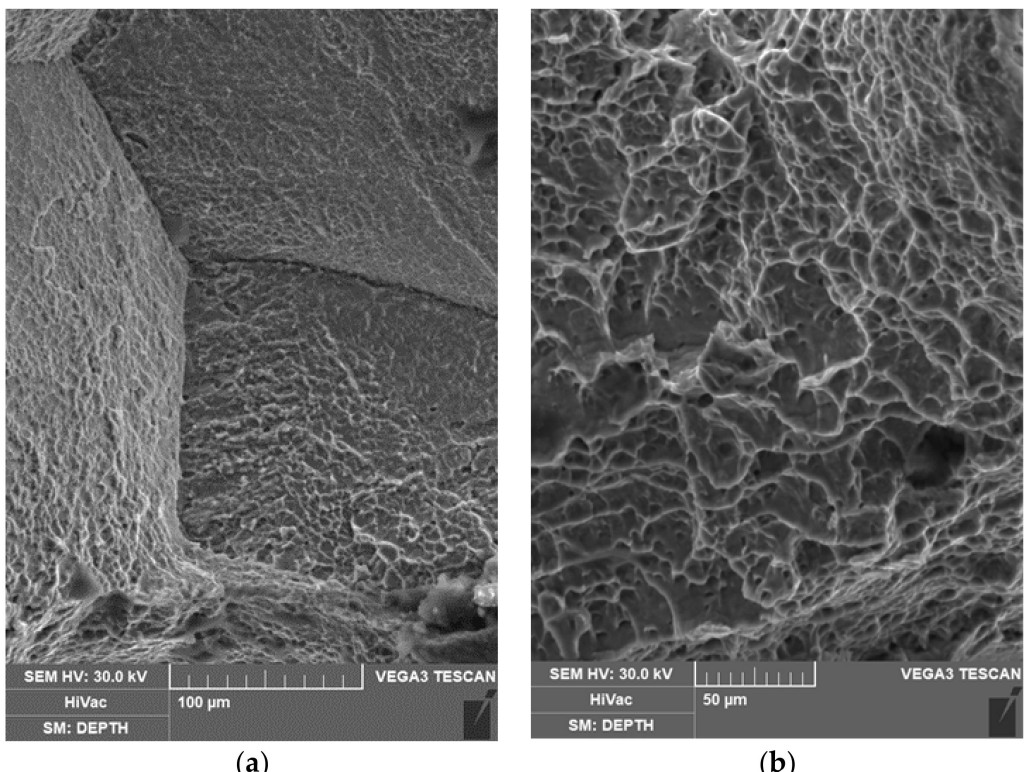

(a)      (b)

**Figure 11.** SEM fractographic analysis of the rupture surface of the sample tested under 275 MPa al low (**a**) and high (**b**) magnification.

### 4. Discussion

The experimental evidence presented in the previous sections (Figure 7), clearly indicated that the T2 treatment results in a microstructure that is more creep-resistant than those produced by subcritical annealing (T1 and R) of the $\alpha'$-martensite. By contrast, T1 and R annealing provides somewhat higher *UTS* at room temperature (1000 vs. 922 MPa). The limited evidence presented in [1] by comparing the alloy in as-deposited state [14] and after T1 treatment already demonstrated that $\alpha'$-martensite, in terms of creep strength, does not substantially differ from equilibrium $\alpha$-phase, a point that seems plainly obvious in Figure 7 (unfortunately, no other data are available in the literature on this subject). Thus, different amounts of $\alpha'$-martensite could not easily explain the significant reduction in creep rates observed for the T2 state in Figure 7.

The first step to rationalize the differences in creep rate could be thus the identification of an appropriate constitutive equation, which is the first point that will be here addressed. Equation (1), as already noted, can be used to successfully describe the minimum creep rate dependence on applied stress for a number of Ti-6Al-4V alloys with different microstructures, including the material investigated here and in [1], in the R and T1 states.

The curve in Figure 7 was thus obtained with *UTS* = 1000 MPa (for the sake of simplicity, the differences among T1, T2 and R states and the underestimation of the *UTS* due to rough surface finishing were neglected). As expected, the description is excellent, except in the very high stress region, where the underestimation of the *UTS* has significant impact on the accuracy of the model.

The problem now arises to properly modify Equations (1)–(3) for describing the results obtained by testing the material in the T2 state. The simplest approach is indeed to follow the same procedure proposed by Barboza et al. [10], by introducing into Equation (2) a $\sigma_0$ term that describes an additional strengthening term. Equation (2) thus becomes

$$\sigma_\rho = (1 - \delta)\sigma - \sigma_0 \tag{4}$$

Equation (4), with $\sigma_0 = 32$ MPa, in combination with Equations (1) and (3), gives the curve for the T2 state in Figure 7. Again, the description is very good, although the real significance of $\sigma_0$, as in the case of the Barboza et al. analysis [10], remains to be clarified. This term, which represents an additional stress required by a dislocation to overcome specific obstacles, was not needed to describe the material in the T1 and R states, where $\sigma_0 = 0$. Thus, the higher creep strength of the alloy after T2 treatment should be related to the presence of a strengthening mechanism not operative after annealing at low temperature.

The experimental evidence presented in the previous Section indicate that two possible sources of strengthening could be operative here: (i) the interaction of dislocations with extensive $\alpha$-$\beta$ interfaces; (ii) the interaction between dislocations and $\alpha_2$-particles. Both these mechanisms could possibly act independently, although prolonged holding at high temperature leads to progressive coarsening of the $\alpha_2$-particles, which should, at least in part, lose their strengthening role. In this sense, as postulated in [10], the role of $\alpha$-$\beta$ interfaces should be predominant under these experimental conditions. If such is the case, one can reasonably conclude that the platelet-like morphology of the $\beta$-phase observed in [1] in the alloy tested in the T1 state is much less effective in obstructing dislocation mobility than the lamellar one presented in Figure 4. The same should apply to bimodal microstructures, in which the lamellar constituent is formed by very large $\alpha$ and $\beta$ lamellae [8,9]. The $\alpha$-$\beta$ interfaces do not provide in this case an effective source of strengthening, being too far apart to act as barriers to dislocation motion. Thus, the scatter observed when comparing the creep response of alloys with nominally similar compositions and Widmanstätten structures observed in [1] could be a direct result of slightly different heat treatments with resulting differences in the spacing of $\alpha$-$\beta$ interfaces and $\alpha_2$-particles.

This preliminary and qualitative conclusion, albeit fully reasonable, obviously needs additional investigations to be confirmed.

## 5. Conclusions

The creep response at 650 °C of a high-temperature annealed Ti-6Al-4V alloy was investigated by constant load experiments. Annealing at 704 or 740 °C provided an experimental creep response very similar to that observed in equiaxed or duplex microstructures. Annealing at 1050 °C, by contrast, significantly enhanced creep resistance. The creep strengthening effect is supposed to be an effect of the interaction dislocations with $\alpha$-$\beta$ interfaces and $\alpha_2$-particles. Thus, the effect can be quantified by introducing an additional strengthening term in the general constitutive equations describing the minimum creep rate dependence on applied stress and temperature.

Given the great interest in additive manufacturing techniques for Ti-6Al-4V alloy, and the lack of data in the literature, more in-depth studies are obviously needed to properly evaluate the effect of strengthening phenomena in terms of creep resistance. Nevertheless, the preliminary evidence obtained in this study suggests that a higher annealing temperature significantly improves creep response at 650 °C by introducing additional obstacles to dislocation motion, in the form of $\alpha$-$\beta$ interfaces and, to a lesser extent, $\alpha_2$ particles.

**Author Contributions:** Conceptualization, S.S.; methodology, S.S. and E.C.; validation, C.P.; formal analysis, C.P. and S.S.; investigation, C.P. and M.C.; resources, S.S. and E.C.; data curation, C.P.; writing—original draft preparation, E.S. and C.P.; writing—review and editing, E.C. and S.S.; visualization, S.S.; supervision, S.S.; project administration, E.C.; funding acquisition, E.S. All authors have read and agreed to the published version of the manuscript.

**Funding:** This research was partially funded by the Grant of Excellence Departments, MIUR-Italy (ARTICOLO 1, COMMI 314–337 LEGGE 232/2016).

**Data Availability Statement:** The data presented in this study are freely available on request from the corresponding author.

**Conflicts of Interest:** The authors declare no conflict of interest.

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
