# Peer review of "Effect of Post-Processing Heat Treatments on Short-Term Creep Response at 650 °C for a Ti-6Al-4V Alloy Produced by Additive Manufacturing"

_metals, doi:10.3390/met12071084_

Round 1
Reviewer 1 Report
The paper is devoted to the problem of studying the nature of the creep of Ti alloy Ti-6Al-4V produced by Selective Laser Melting (SLM). The work makes a good impression, but there are some questions and comments to the paper text.
1. The authors should specify the chemical composition of the powders used, present the data on the particle size distribution, and also indicate the manufacturer of the powders. Information should also be provided on the chemical composition of the alloys, with particular emphasis on the oxygen content.
2. On what platform (substrate) did the samples fuse? Was platform heating used?
3. In what inert medium was the samples fabricated by SLM?
4. What was the porosity of the fabricated samples?
5. Please specify the equipment, which the samples were annealed in and the uncertainty of the temperature measurements. Also, please indicate the average cooling rates of the samples.
6. Were the samples heated together with the furnace or placed in a heated-up furnace?
7. Have the surfaces of the samples been pre-treated?
8. How many identical samples were made total? How many samples were tested in the states R, T1, and T2?
9. In the last paragraph before Fig. 1, the authors wrote that Fig. 1 presents an image of a specimen for tensile testing. The caption to Fig. 1 indicates that this is a specimen for creep testing. Did the authors use the same specimens for both creep and tensile tests?
10. In the methodological section of the paper, the authors should indicate the stresses, which the creep tests were carried out at.
11. The tension curves for the samples R, T1, and T2 must be presented.
12. The authors should present the results of the X-ray diffraction phase analysis (XRD) of the samples and evaluate the phase composition of the samples quantitatively. Without this critically important information, any correct analysis of the results obtained is extremely problematic. The XRD results will make it possible to supplement the TEM data and characterize fully the microstructure of the alloys.
13. What is the HRC hardness of the samples in R, T1, and T2 conditions?
14. It would be helpful to present the creep curves e(t) and indicate the sections, which the minimum creep rate was determined from.
15. Has the creep activation energy been evaluated?
16. Figure 6 should be moved to Subsection 3.2 and described in details.
17. The authors should show the measurement errors in figures 6 and 7.
18. The conclusion drawn about the nonmonotonic nature of the change in hardness with the time of creep testing seems unreliable to the reviewer - the scale of changes in hardness is very small. If one plots the hardness measurement error (0.5-1 HRC) standard for the SLM materials in Fig. 7, then almost all experimental points will fall within the confidence interval and will be close to the hardness for the T2 material.
19. Why the data on the dependence of hardness on the test time for other materials (R, T1) is not presented in the paper?
20. The HRC values shown in Fig. 7 are abnormally high for the Ti6Al4V alloy. What was the depth of the indentations when measuring the hardness? The thickness of the sample was 0.5 mm only and usually the Rockwell hardness HRC is not measured on such thin samples.
21. The samples were tested in air for a long time at high temperatures. This always leads to oxidation of the metal and to the saturation of the surface layers of the samples with oxygen. What was the oxygen concentration in the samples after the creep tests?
22. The authors should present the results of fractographic analysis of specimen fractures. Without these data, the conclusion about the effect of microstructure on the nature of creep is not complete.
23. To describe the results obtained, the authors introduced the threshold limit stress s0, which the authors call “additional stress” equal to 32 MPa into equation (4). This assumption seems reasonable but it should be noted that the value of the threshold limit stress should be different for materials R, T1, and T2. It would be helpful to analyze the results assuming the annealing to affect the value of s0.
24. Can the authors compare the values of the minimum creep rate obtained in their work with the data reported by other authors?
After addressing the comments made, the article may be published in "Metals".
Reviewer 2 Report
Manuscript numbered “metals-1764700” has been reviewed:
The introduction needs some improvements.
Please use laser beam powder bed fusion (LB-PBF) instead of selective laser melting (SLM).
Results have been just reported, please compare your finding with other research.
Please add the Ti-6Al-4V heat treatment diagram and phase formation on paper.
Please add process parameters and chemical composition in tables.
Fallowing papers are suggested for the introduction and result section:
A study on surface morphology and tension in laser powder bed fusion of Ti-6Al-4V
Study on the influence of process parameters on high-performance Ti-6Al-4V parts in laser powder bed fusion
Three-dimensional printing technologies for dental prosthesis: a review
Optimisation of single contour strategy in selective laser melting of Ti-6Al-4V lattices
Study of anisotropy through microscopy, internal friction and electrical resistivity measurements of Ti-6Al-4V samples fabricated by selective laser melting
Round 2
Reviewer 1 Report
The reviewer hasn't comments on the article.